# Multi-task gradient boosting with multi-modal molecular representations for simultaneous prediction of drug clearance and volume of distribution

Gyu-Seong Lee[1‡], Hyeong-Gyu Choi[1‡], Dae-Keun Park[1]*, Hyo Young Kim[2]*

1 Department of AI Healthcare Convergence, Graduate School, CHA University, Pocheon-si, Republic of Korea, 2 Department of Artificial Intelligence, Tech University of Korea, Siheung-si, Republic of Korea

‡ These authors contributed equally to this work.
* dkpark@cha.ac.kr (DKP); hyoyoung@tukorea.ac.kr (HYK)

## Abstract

Accurate prediction of pharmacokinetic parameters—particularly drug clearance (CL) and volume of distribution (VD)—is critical in early drug development, yet existing approaches largely treat these physiologically correlated endpoints as independent tasks and rely on single molecular data modalities. To address these gaps, we propose a Multi-Task Gradient Boosting Machine (MTGBM) framework that simultaneously predicts CL and VD by integrating three complementary molecular modalities—CNN-based structural embeddings, MLP-derived descriptor embeddings, and physicochemical and preclinical PK parameters—into shared decision trees. MTGBM outperformed single-task LightGBM baselines in MSE and $R^2$ for both targets, with CL GMFE also showing improvement, except for VD GMFE in the low VD range. These findings were further reinforced by 10 repeated random splits of the entire dataset. SHAP analysis revealed complementary contributions from all three modalities and suggested cross-target information sharing between CL and VD embeddings. These results establish MTGBM as a proof-of-concept for multi-modal, multi-task pharmacokinetic modeling, offering a foundation for future integration with larger datasets or foundation model-derived representations.

## 1. Introduction

Accurate prediction of pharmacokinetic (PK) parameters is crucial for successful drug development, as these parameters directly influence drug efficacy, safety, and dosing regimens. Among the key PK parameters, drug clearance (CL) and volume of distribution (VD) critically determine drug exposure and half-life, essential for early drug discovery. Despite significant advances in computational methods, reliable prediction of these parameters remains challenging due to the complex interplay of molecular properties, physiological factors, and drug-target interactions.

**Data availability statement:** The data used in this study were obtained from a previously published study by Iwata et al. (2022). Specifically, we used the dataset of 700 pharmaceutical compounds curated and reported in that study. The original data are publicly available as described in the following publication: Iwata, H., Matsuo, T., Mamada, H., Motomura, T., Matsushita, M., Fujiwara, T., Maeda, K., & Handa, K. (2022). Predicting Total Drug Clearance and Volumes of Distribution Using the Machine Learning-Mediated Multimodal Method through the Imputation of Various Nonclinical Data. Journal of Chemical Information and Modeling, 62(17), 4057–4065. No new data were generated for this study.

**Funding:** This work was supported by the Technology Innovation Program (RS-2025-11902968, Development of AI-Based Formulation Optimization Technology for Subcutaneous Conversion of Intravenous Antibody Drugs) funded By the Ministry of Trade, Industry & Resources (MOTIR, Korea).

**Competing interests:** The authors have declared that no competing interests exist.

Traditional approaches to predicting CL and VD have primarily relied on single-task models that treat each parameter independently, utilizing either quantitative structure-activity relationship (QSAR) models or machine learning algorithms trained on molecular descriptors. However, these conventional methods often fail to capture the inherent correlations between PK parameters and struggle to leverage the full spectrum of available molecular information. Moreover, most existing models are limited by their reliance on a single data modality, typically focusing on either numerical molecular descriptors or structural representations, thereby missing potentially valuable complementary information encoded in different data formats.

Recent advances in multi-modal and multi-task learning present new opportunities to overcome these limitations. Multi-modal approaches can integrate diverse data representations including molecular structures, SMILES strings, and physicochemical properties to create more comprehensive molecular representations. Similarly, multi-task learning can exploit the shared underlying patterns between related PK parameters, potentially improving prediction accuracy through joint learning of correlated tasks. Despite these promising developments, the application of combined multi-modal and multi-task frameworks to PK parameter prediction remains largely unexplored.

In this study, we propose and evaluate a Multi-Task Gradient Boosting Machine (MTGBM) framework for the simultaneous prediction of CL and VD, building on the MT-GBM formulation introduced by Ying et al. [1]. Rather than introducing a new optimization paradigm, this work represents an application-level contribution that systematically explores how multi-modal molecular representations and multi-task gradient boosting can be combined for pharmacokinetic modeling—a combination that, to our knowledge, has not been previously evaluated. Our approach integrates multiple data modalities including molecular images, SMILES representations, and numerical descriptors using deep learning-based embedding techniques, while employing a shared latent structure to capture the intrinsic relationships between CL and VD. Using a curated dataset of approximately 700 drug compounds originally compiled by Iwata et al. [2] and further enriched with additional structural information obtained from PubChem and DrugBank, we evaluate the practical utility of this integration and assess whether the multi-task framework yields consistent predictive improvements over single-task baselines.

This study makes three primary contributions at the application level:

(1) We present a systematic evaluation of combining multi-modal molecular representations—including CNN-based structural embeddings, molecular fingerprints, and physicochemical descriptors—within a multi-task gradient boosting framework for simultaneous CL and VD prediction.

(2) We employ BorutaSHAP for robust feature selection and assess the stability of model performance through repeated random split analysis, providing a more rigorous empirical evaluation than a single train-test split.

(3) We enhance model interpretability using SHAP-based analysis, identifying key molecular features influencing PK behavior.

Our results serve as a proof-of-concept demonstrating the feasibility of integrating multi-modal molecular representations with multi-task gradient boosting for pharmacokinetic prediction, offering a foundation for future refinement and potential extension to broader ADME modeling applications.

## 2. Literature review

### 2.1. Traditional and machine learning-based PK prediction

The prediction of pharmacokinetic parameters has evolved from traditional empirical methods to sophisticated machine learning approaches. Early methods relied on allometric scaling and in vitro-in vivo extrapolation (IVIVE), with Ring et al. [3] demonstrating through the PhRMA consortium that these approaches achieved predictions within 2-fold for only 50–60% of compounds. Traditional methods showed systematic biases, particularly underpredicting clearance for low-clearance compounds [4]. Berellini and Lombardo [5] improved upon mechanistic approaches by refining the Øie-Tozer equation with experimentally determined parameters, achieving comparable accuracy while avoiding animal testing.

The machine learning revolution has transformed PK prediction by capturing complex non-linear relationships. Danishuddin et al. [6] documented a decade of ML progress in PK modeling, highlighting advantages in handling large descriptor sets despite challenges in interpretability. Recent comparative studies demonstrate ML's superiority: Kosugi and Hosea [7] showed ML models matching or exceeding IVIVE accuracy, while Keefer et al. [8] reported an ML-IVIVE model achieving 2.5-fold average error across 645 drugs, outperforming mechanistic models. For volume of distribution, Murad et al. [9] found direct ML models achieved 75% of predictions within 3-fold error on 950 clinical compounds, demonstrating that data-driven approaches can match traditional methods when sufficient training data is available. More recently, Li et al. [10] demonstrated the potential of hybrid approaches by combining ML with PBPK modeling, achieving improved predictions while maintaining mechanistic interpretability.

### 2.2. Multi-modal and multi-task learning in drug discovery

Multi-modal learning has emerged as a powerful strategy for enhancing prediction accuracy by integrating diverse molecular representations. Iwata et al. [11] pioneered multimodal PK prediction by combining rat clearance data with chemical descriptors, achieving GMFE of 2.68 compared to conventional single-modal approaches. Their subsequent work [2] extended this framework to predict both CL and VD within a single multimodal framework, though each target was predicted independently. By leveraging missing-value imputation, they achieved accuracy comparable to animal scaling (GMFE ~1.9–1.6) while eliminating animal experiments.

Multi-task learning offers complementary advantages by exploiting correlations between related endpoints. Demir-Kavuk et al. [12] demonstrated early success with DemQSAR for simultaneous CL and VD prediction, achieving competitive accuracy with improved interpretability. The theoretical foundation was significantly advanced by Ying et al. [1], who introduced MT-GBM with shared decision trees that optimize multiple targets simultaneously. Their framework showed consistent improvements when learning related tasks together, with recent applications by Wang et al. [13] in materials science confirming its versatility for multi-property optimization problems. Despite these parallel advances, multi-modal and multi-task learning have largely been pursued independently in the context of PK prediction, leaving their synergistic combination largely unexplored.

More recently, large-scale pretraining and foundation model approaches have emerged as a powerful paradigm for molecular property prediction, including ADME-relevant endpoints. Bang et al. [14] demonstrated that pretraining molecular foundation models across 21 sequential ADME endpoints substantially improved downstream drug-likeness prediction, with performance gains of up to 18.2% over task-specific baselines. Similarly, Cai et al. [15] introduced ChemFM, a 3-billion-parameter foundation model pretrained on 178 million molecules via self-supervised learning, achieving directional improvements across 34 molecular property prediction benchmarks including ADMET endpoints. These studies

highlight that large-scale multi-task pretraining can yield generalizable molecular representations that outperform models trained on limited, task-specific datasets, particularly in low-data regimes.

## 2.3. Research gaps and our contribution

Current limitations include: (1) independent modeling of physiologically related parameters, (2) multi-modal methods restricted to single tasks, and (3) multi-task approaches not leveraging diverse molecular representations. Recent industry perspectives, such as those by Gawehn et al. [16] and Bassani et al. [17], emphasize the need for integrated frameworks combining mechanistic understanding with advanced ML techniques.

Our MTGBM framework serves as a proof-of-concept for systematically combining multi-modal representations with multi-task learning for simultaneous CL and VD prediction. Building on MT-GBM's theoretical foundation [1] and successful multi-modal PK strategies [2], this work evaluates whether leveraging shared latent structures and complementary information across modalities yields practical predictive benefits. Rather than proposing a new algorithmic framework, this study provides an empirical evaluation of this combination and offers interpretable insights through SHAP analysis, serving as an application-level proof-of-concept for multi-modal, multi-task approaches in pharmacokinetic modeling.

Unlike large-scale pretraining approaches that require millions of molecules and substantial computational resources, our framework operates in a constrained low-data setting (~700 compounds) where task-specific supervised embeddings offer a practical alternative. While foundation model-based representations may yield stronger generalization, exploring whether the MTGBM multi-task boosting layer can serve as a lightweight downstream learner on top of such pretrained embeddings remains a promising direction for future work.

## 3. Methods

### 3.1. Dataset

We utilized the publicly available dataset from Iwata et al. [2] containing pharmacokinetic parameters for approximately 700 drug compounds. The dataset comprises four sheets: CL imputed training dataset, CL evaluation dataset, VDss imputed training dataset, and VDss evaluation dataset. Each contains human pharmacokinetic parameters alongside preclinical variables including rat, dog, and monkey PK values, physicochemical properties (pKa), and in vitro data (plasma protein binding). We merged CL and VD datasets based on molecular structure (SMILES) and compound names after removing whitespace, retaining only compounds with both measurements to enable multi-task learning. As this study utilized only publicly available datasets and did not involve human subjects, animal experiments, or clinical interventions, ethical approval and informed consent were not required.

### 3.2. Data preprocessing and quality control

Data preprocessing began with removing compounds exceeding physiological ranges (human VDss ≥ 5 L/kg or CL ≥ 20 mL/min/kg); these thresholds represent physiological upper limits, as values exceeding these ranges are considered biologically implausible and likely represent data entry errors or experimental artifacts, followed by winsorization at the 1st and 99th percentiles for all numeric features. For normalization, we excluded variables with prefix 'Caco_2' or 'water_solubility' (Caco_2 permeability data were excluded due to their categorical nature in the dataset, while water_solubility measurements showed highly irregular distributions with excessive skewness that could not be adequately normalized), then applied log1p transformation and standardization to target variables and animal PK parameters, while only standardizing physicochemical properties. All scaling parameters were preserved for inverse transformation during evaluation.

### 3.3. Multi-modal feature extraction

**3.3.1. Molecular image embeddings via CNN.** We generated molecular structure images from SMILES strings using RDKit's molecular drawing functionality, creating standardized 224 × 224 pixel RGB images. These images were

 

processed through a modified ResNet18 architecture [18,19], where we retained the backbone up to global average pooling, followed by a custom head consisting of a fully connected layer (512→2 dimensions), dropout regularization, and a final regression layer. The 2-dimensional embedding layer used linear activation to preserve the full range of learned representations.

We trained separate CNN models for CL and VD predictions following a three-step protocol. Throughout all stages, the test set was reserved exclusively for final embedding extraction and was not involved in any model fitting or hyperparameter selection:

(1) Hyperparameter optimization using Optuna with 100 trials, evaluating candidate configurations on the validation set with early stopping, exploring learning rates (1e-5 to 1e-2), dropout rates (0.0–0.5), batch sizes (8, 16, 32, 64), optimizers (Adam, SGD, RMSprop), and epochs (50–1000)

(2) Final model training on combined train+validation data using best hyperparameters, with early stopping to select the best model weights

(3) Embedding extraction by applying the best model in inference mode to each data split independently, producing 2-dimensional representations without further parameter updates.

**3.3.2. Molecular descriptor embeddings via MLP.** We computed molecular features by combining RDKit descriptors (molecular weight, LogP, hydrogen bond donors/acceptors, TPSA), Morgan fingerprints (radius = 2, 256 bits), and Mol2Vec embeddings from a pre-trained 300-dimensional model. These heterogeneous features were concatenated and processed through multi-layer perceptrons optimized via Optuna with 100 trials.

Following the same three-step protocol as CNN training, with the test set strictly reserved for final embedding extraction, we optimized MLP architectures exploring 1–3 hidden layers, 32–256 units per layer, dropout rates (0.0–0.5), learning rates (1e-4 to 1e-2), and optimizers (Adam, RMSprop). Hyperparameter selection was based solely on validation set performance. For the final model, training was conducted on combined train+validation data, after which 2-dimensional embeddings were extracted from a bottleneck layer with linear activation positioned immediately before the final regression output, applied to all data splits including the test set using the trained model in inference mode. The final architectures were determined through hyperparameter optimization, with different configurations selected for CL and VD based on validation performance.

## 3.4. Feature selection via BorutaSHAP

We employed BorutaSHAP, combining the Boruta algorithm with SHAP values for robust feature selection. The algorithm creates shadow features through random permutation, trains a LightGBM model on both original and shadow features, and selects features whose SHAP-based importance consistently exceeds the maximum shadow feature importance. We configured BorutaSHAP with LightGBM (max_depth = 5, GPU acceleration), 5,000 iterations, and 80th percentile threshold for shadow comparison. Feature selection was performed independently for each target using training data. Based on the analysis, BorutaSHAP identified 11 optimal features each for CL and VD prediction. For multi-task modeling, we used the union of features selected for both CL and VD.

## 3.5. Model development

**3.5.1. Single-task baseline models.** Single-task LightGBM models served as baselines for evaluating our multi-task approach. We performed hyperparameter optimization using Optuna with 100 trials, exploring learning rates (0.005–0.05), number of leaves (8–64), max depth (6–15), and regularization parameters. The optimization targeted GMFE on the validation set. Final models were trained with the best hyperparameters, employing GPU acceleration with early stopping (100–150 rounds patience) monitoring validation MSE, allowing up to 15,000 boosting rounds. Separate models were trained for CL and VD using their respective selected features.

### 3.5.2. Multi-task gradient boosting machine (MTGBM).

Our MTGBM implementation extends gradient boosting to predict multiple targets through shared tree structures, following the approach proposed by Ying et al. [1]. We developed a custom objective function computing gradients and Hessians for both targets simultaneously, returning four arrays to accommodate LightGBM's optimization routines. Due to computational constraints, we used estimated optimal parameters based on single-task results: learning rate 0.014, 44 leaves, max depth 13, with L1/L2 regularization (0.71/2.89). MTGBM was configured with objective = "custom", num_labels=2, tree_learner="serial2", using CPU processing. The shared tree structure allows each leaf to predict both CL and VD, enabling joint pattern learning. The feature set comprised the union of individually selected features.

## 3.6. Model training and evaluation

Dataset splitting was performed using a robust optimization algorithm evaluating 10,000 random splits to identify the optimal 60/20/20 partition based on distribution similarity metrics. Hyperparameter optimization for single-task models employed Optuna with 100 trials using TPE sampling, optimizing for GMFE on the validation set. The search space included learning rate (0.005–0.05 log scale), leaves (8–64), depth (6–15), min_child_samples (10–50), and regularization parameters. For MTGBM, we used direct training with estimated parameters due to computational constraints. Final models were evaluated on the held-out test set with early stopping (100–150 rounds) monitoring performance.

To assess the stability of results across different data partitions, we conducted 10 repeated random splits of the preprocessed dataset, each using an independent 60/20/20 partition generated with a different random seed. For each split, the complete modeling pipeline was executed independently, including normalization, CNN and MLP embedding generation, BorutaSHAP feature selection, LightGBM hyperparameter optimization, and MTGBM training with fixed estimated parameters. Model performance was evaluated on each held-out test set, and results are reported as mean ± standard deviation across all 10 runs. The directional consistency of MTGBM versus LightGBM performance was additionally assessed by counting the number of runs in which MTGBM achieved lower MSE than LightGBM for each target.

To ensure rigorous evaluation, the test set was maintained as a strictly held-out set throughout the entire modeling pipeline, as described in Sections 3.3.1 and 3.3.2. This end-to-end separation ensures that reported test set metrics reflect unbiased estimates of generalization performance.

## 3.7. Performance metrics and statistical analysis

Performance was evaluated using MSE on log1p-transformed scale and Geometric Mean Fold Error (GMFE) on original scale. GMFE provides a symmetric fold-error measure that equally penalizes over- and under-predictions, making it particularly relevant for pharmacokinetic predictions where both overestimation and underestimation of drug exposure can have clinical consequences. Statistical significance was assessed using the Diebold-Mariano test with MSE criterion and h = 1. For interpretability, we generated SHAP values using TreeExplainer for single-task models and KernelExplainer with reduced sample sizes (200 test samples, 30 background samples) for computational efficiency with MTGBM. All tests used $\alpha = 0.05$ significance level with two-tailed tests where applicable. For the repeated random split analysis, model performance across 10 runs is reported as mean ± standard deviation of MSE for each target. The directional consistency of MTGBM relative to LightGBM was quantified as the proportion of runs in which MTGBM achieved lower test set MSE, referred to as the win rate.

## 4. Results

### 4.1. Model training and optimization performance

The hyperparameter optimization process for single-task LightGBM models completed successfully using Optuna with 100 trials per target. For CL prediction, the optimal configuration achieved a validation GMFE of 1.537 (learning rate: 0.030, leaves: 19, max depth: 6). The VD model optimization yielded a validation GMFE of 1.432 with learning rate 0.037, 52

leaves, and max depth 6. Both models employed early stopping mechanisms (patience ≈150 rounds) based on validation performance, effectively preventing overfitting and ensuring stable convergence.

The MTGBM model was trained using estimated optimal hyperparameters derived from single-task LightGBM results, showing stable convergence without negative transfer between tasks. Final validation metrics demonstrated balanced performance across both targets without signs of negative transfer learning.

## 4.2. Feature selection analysis

BorutaSHAP analysis indicated the importance and relevance of the selected features, with 6 overlapping features between CL and VD predictions, as summarized in Table 1.

Fig 1 illustrates the BorutaSHAP selection process for CL prediction, showing clear separation among accepted features (importance exceeding shadow feature distribution), rejected features (below shadow distribution), and shadow features (randomized reference distribution). Similarly, Fig 2 presents the corresponding BorutaSHAP analysis for VD prediction, where the selected features spanned multiple modalities: 4 physicochemical/in vitro properties (pKa_Acid, rat_fup, human_fup, dog_fup), 4 preclinical PK parameters (rat_VDss_L_kg, dog_CL_mL_min_kg, dog_VDss_L_kg, monkey_VDss_L_kg), and 3 learned embeddings (cnn_vec1_CL, mlp_vec1_VD, mlp_vec2_VD) out of 11 total selected features. This distribution across modalities suggests that no single data type dominates the prediction, supporting the rationale for adopting a multi-modal feature set.

## 4.3. Predictive performance comparison

Comprehensive evaluation on the test set (n = 139) revealed statistically significant improvements for MTGBM over single-task LightGBM models, as detailed in Table 2. For CL prediction, MTGBM achieved an MSE of 14.580 compared to LightGBM's 19.081, representing a 23.6% improvement. The coefficient of determination increased from 0.140 to 0.196 (+40.0%), and GMFE also decreased from 2.628 to 2.306 (−12.3%), indicating consistent improvements across all three metrics for CL prediction.

For VD prediction, MTGBM demonstrated substantial improvements with MSE decreasing from 1.130 to 0.757 (−33.0%) and R² increasing from 0.212 to 0.418 (+97.2%). The GMFE, however, increased from 2.014 to 2.286 (+13.5%), indicating higher fold errors despite the sizable gains in MSE and R². This divergence — observed only for VD — warrants careful interpretation and is examined in detail via range-stratified analysis below.

Fig 3 presents the prediction scatter plots for single-task LightGBM models across all data splits, with the dashed lines indicating 2-fold error boundaries. In comparison, Fig 4 shows the corresponding scatter plots for MTGBM, where the predictions show generally comparable distributions to LightGBM with modest improvements in R², particularly for VD prediction in the test set.

To investigate the divergence between MSE and GMFE observed for VD prediction, a range-stratified analysis was conducted by partitioning test set compounds into three VD value ranges. As summarized in Table 3, MTGBM demonstrated superior GMFE in both the mid (0.5–2.0 L/kg, GMFE: 1.446 vs 1.740, Δ = −0.294) and high (>2.0 L/kg, GMFE: 2.043 vs 2.467, Δ = −0.424) VD ranges. However, in the low VD range (<0.5 L/kg, n = 42), MTGBM GMFE increased

**Table 1. Features selected by BorutaSHAP for each prediction target.**

| Target | Selected Features | Count |
|--------|-------------------|-------|
| CL | pKa_Acid, mlp_vec2_CL, pKa_base, rat_fup, rat_CL_mL_min_kg, human_fup, cnn_vec2_CL, dog_CL_mL_min_kg, mlp_vec1_CL, mlp_vec1_VD, monkey_VDss_L_kg | 11 |
| VD | pKa_Acid, cnn_vec1_CL, rat_fup, human_fup, rat_VDss_L_kg, dog_fup, dog_CL_mL_min_kg, dog_VDss_L_kg, mlp_vec1_VD, monkey_VDss_L_kg, mlp_vec2_VD | 11 |
| Union | All unique features from both targets | 16 |

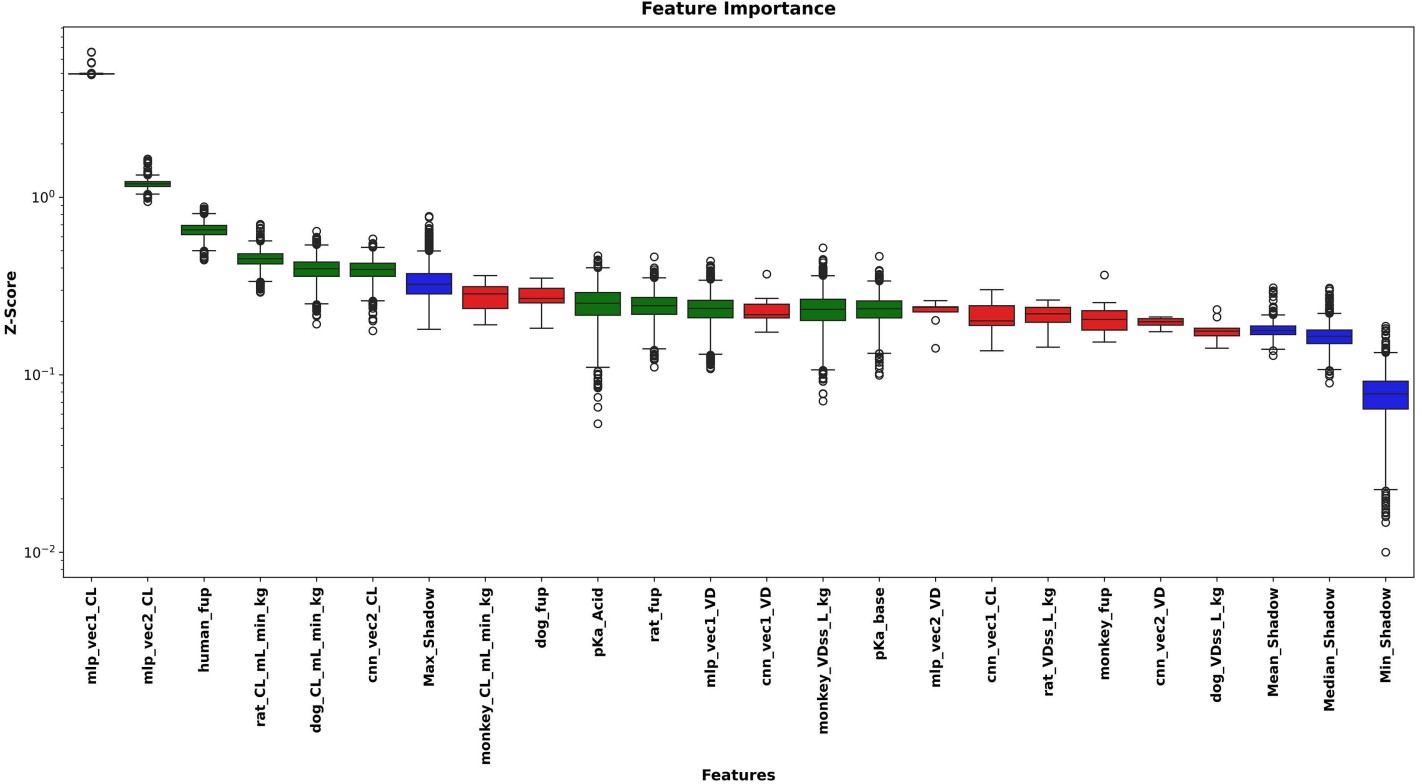

**Fig 1. BorutaSHAP feature importance for CL prediction.** Boxplots show the distribution of SHAP-based importance scores (Z-score, log scale) across 5,000 iterations for each candidate feature. Green boxes indicate accepted features; red boxes indicate rejected features; blue boxes represent Max, Mean, Median, and Min Shadow reference distributions. Features are ordered by median importance from left to right.

relative to LightGBM (4.416 vs 2.142, Δ=+2.274), indicating that the overall GMFE deterioration was driven predominantly by compounds in this range. Fig 5 presents the range-stratified GMFE comparison and the corresponding differences, illustrating that the overall GMFE deterioration is driven exclusively by the low VD range.

### 4.4. Statistical significance assessment

The Diebold-Mariano test evaluated the statistical significance of performance differences between models. For CL prediction, the test yielded DM statistic=2.370 (p=0.019), while VD prediction showed DM statistic=2.898 (p=0.004). Both targets reached statistical significance at the α=0.05 level, confirming that MTGBM's MSE improvements over LightGBM are unlikely to reflect chance variation. Fig 6 illustrates the comparative performance through multiple visualizations, including MSE comparison bars, prediction scatter plots, and residual analyses, confirming the absence of systematic prediction biases in either model.

To further assess the stability of these results, we conducted 10 repeated random splits and evaluated both models on each independent test set. For CL prediction, MTGBM achieved lower MSE than LightGBM in 8 out of 10 runs (win rate: 80%), with mean MSE of 16.663±3.373 versus 19.769±3.251 for LightGBM. For VD prediction, MTGBM demonstrated consistent superiority across all repeated splits, achieving lower MSE in all 10 runs (win rate: 100%), with mean MSE of 0.669±0.133 versus 0.816±0.167 for LightGBM. Notably, when MTGBM underperformed in CL prediction, the margin of difference was relatively modest (mean Δ=+1.52), whereas MTGBM's wins were associated with substantially larger improvements (mean Δ=−4.26), suggesting a pronounced asymmetric performance pattern in which MTGBM's

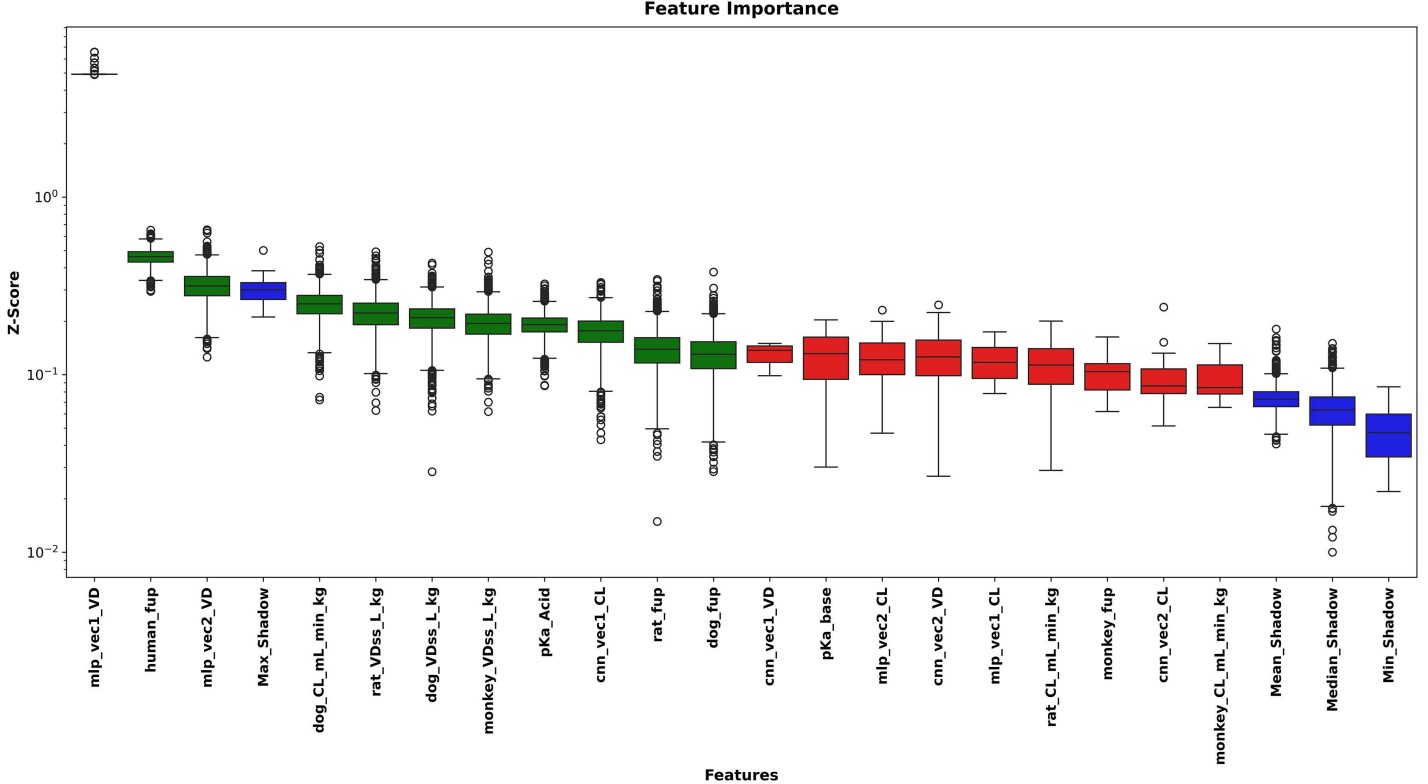

**Fig 2. BorutaSHAP feature importance for VD prediction.** Boxplots show the distribution of SHAP-based importance scores (Z-score, log scale) across 5,000 iterations. Color coding follows the same convention as Fig 1.

**Table 2. Test set performance comparison between single-task LightGBM and multi-task MTGBM models.**

| Model | Target | MSE | GMFE | R² | MSE Improvement |
|---|---|---|---|---|---|
| LightGBM | CL | 19.081 | 2.628 | 0.140 | – |
| MTGBM | CL | 14.580 | 2.306 | 0.196 | −23.6% |
| LightGBM | VD | 1.130 | 2.014 | 0.212 | – |
| MTGBM | VD | 0.757 | 2.286 | 0.418 | −33.0% |

gains considerably outweigh its losses. These results indicate that the improvements observed in the primary single-split analysis are not an artifact of a particularly favorable data partition, but rather reflect a consistent directional tendency of MTGBM toward improved performance — most notably for VD prediction, where MTGBM's advantage was consistent across all 10 independent data partitions, suggesting particularly robust benefits. The asymmetric pattern observed in CL — where gains substantially outweigh losses — further supports the practical utility of the multi-task framework overall. It is worth noting that MTGBM's CL standard deviation was slightly higher than LightGBM's (3.373 vs 3.251), which may reflect the use of estimated rather than fully optimized hyperparameters, as discussed in the limitations. Figs 7 and 8 illustrate the per-run MSE differences (MTGBM−LightGBM) for CL and VD respectively, visually confirming the directional consistency of MTGBM's improvements across independent data partitions.

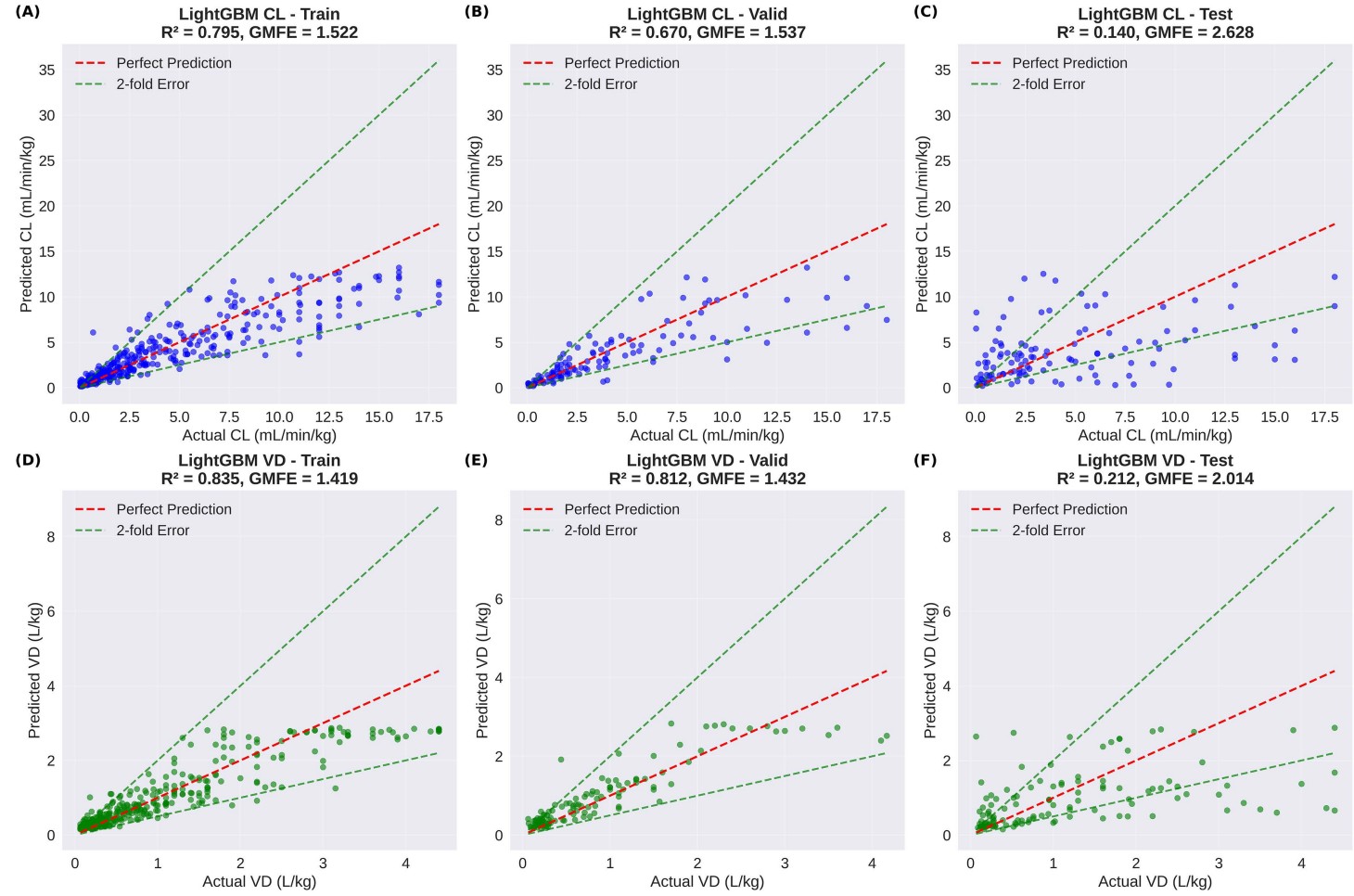

**Fig 3. LightGBM prediction scatter plots across train, validation, and test sets. (A)** CL train, **(B)** CL validation, **(C)** CL test, **(D)** VD train, **(E)** VD validation, and **(F)** VD test predictions. Red dashed lines indicate perfect prediction; green dashed lines indicate 2-fold error boundaries. R² and GMFE are reported for each split.

### 4.5. Model interpretability analysis

SHAP value analysis revealed distinct importance patterns in the MTGBM model, providing insights into how multi-task learning leverages shared representations.

Fig 9 presents the SHAP summary plots for MTGBM, highlighting the top 10 contributors for each target. For CL prediction, MLP embeddings dominated with mlp_vec1_CL showing the highest mean |SHAP value|, followed by mlp_vec2_CL, with preclinical PK parameters (dog_CL_mL_min_kg, dog_VDss_L_kg, rat_CL_mL_min_kg) and human_fup constituting the next tier. Notably, VD-specific embeddings (mlp_vec1_VD, mlp_vec2_VD) also appeared among the top 10 CL predictors, consistent with cross-target information sharing and suggesting the potential of the multi-task framework to leverage correlations between pharmacokinetic parameters. However, since these embeddings were trained using the respective target variables, their prominence in SHAP analysis may partly reflect the supervised nature of the representations rather than solely the effect of multi-task learning.

For VD prediction, preclinical PK parameters dominated — with dog_VDss_L_kg as the primary contributor (33.5% of top-10 SHAP importance) — followed by the MLP embedding mlp_vec1_VD (17.0%) and physicochemical descriptors

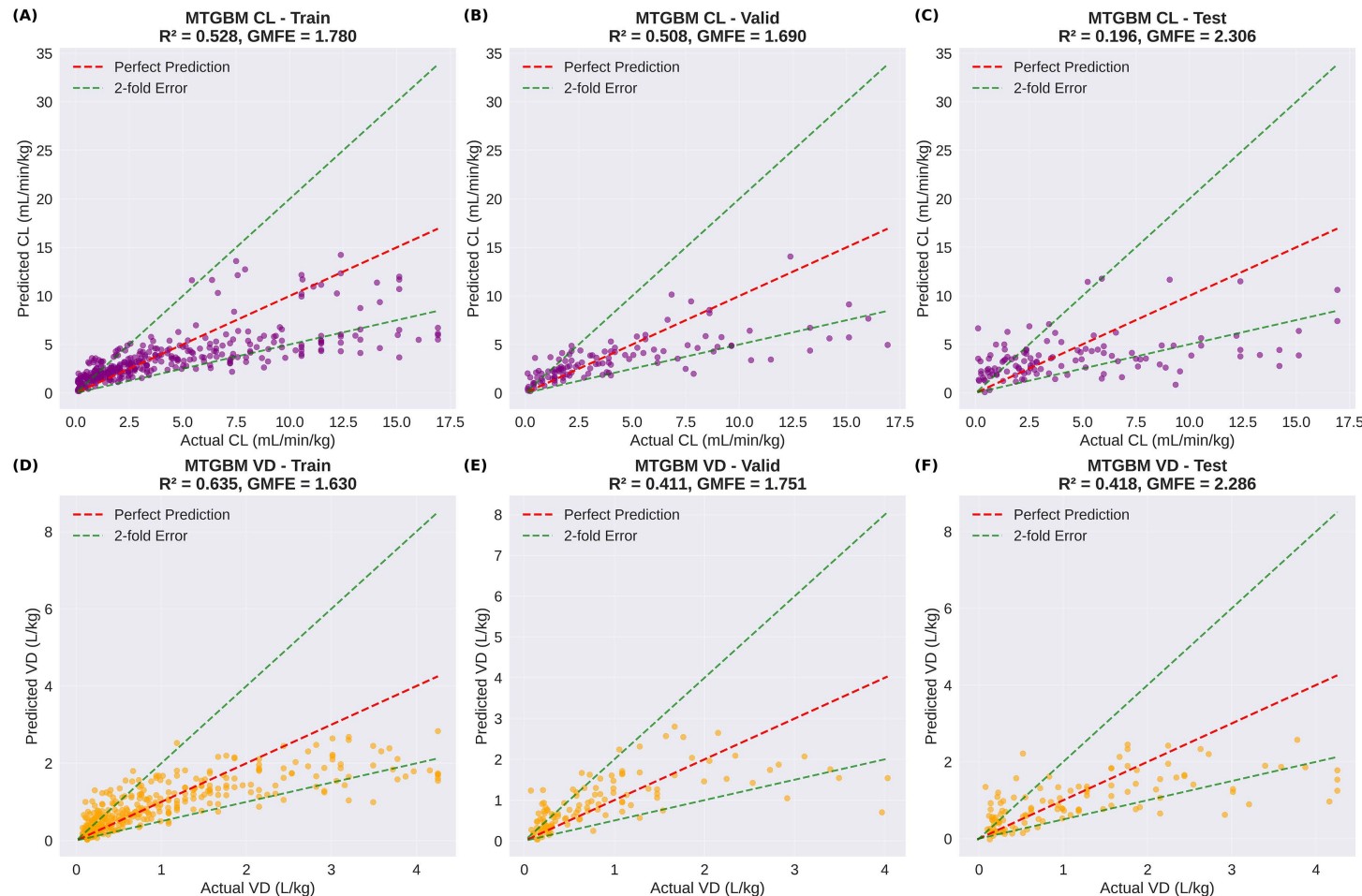

**Fig 4. MTGBM prediction scatter plots across train, validation, and test sets. (A)** CL train, **(B)** CL validation, **(C)** CL test, **(D)** VD train, **(E)** VD validation, and **(F)** VD test predictions. Red dashed lines indicate perfect prediction; green dashed lines indicate 2-fold error boundaries. R² and GMFE are reported for each split.

**Table 3. Range-stratified GMFE comparison between LightGBM and MTGBM on the VD test set.**

| VD Range | N | LightGBM GMFE | MTGBM GMFE | Δ (MT−LGB) |
|---|---|---|---|---|
| Low (<0.5 L/kg) | 42 | 2.142 | 4.416 | +2.274 |
| Mid (0.5–2.0 L/kg) | 51 | 1.740 | 1.446 | −0.294 |
| High (>2.0 L/kg) | 24 | 2.467 | 2.043 | −0.424 |

Note: Range-stratified analysis was conducted on the 117 compounds with available VD measurements, out of the 139 total test set compounds.

(pKa_base, pKa_Acid, human_fup, rat_fup). The modality distribution of top-10 contributors (preclinical PK: 47.8%, MLP embeddings: 28.0%, physicochemical properties: 24.2%) supports the rationale for our multi-modal approach, suggesting that different information sources capture complementary aspects of drug distribution. Notably, the prominence of dog_VDss_L_kg as the leading predictor of human VD is biologically consistent with established allometric scaling principles, where interspecies VDss is widely regarded as one of the most reliable empirical predictors of human distribution volume.

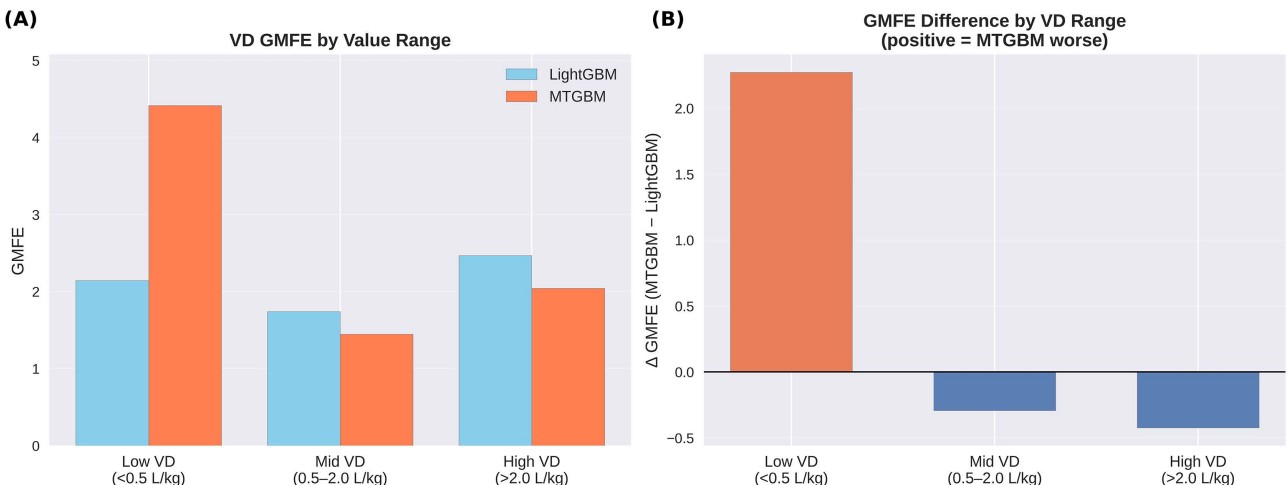

**Fig 5. Range-stratified GMFE comparison for VD prediction. (A)** GMFE values for LightGBM and MTGBM across three VD ranges: low (<0.5 L/kg), mid (0.5–2.0 L/kg), and high (>2.0 L/kg). **(B)** Δ GMFE (MTGBM−LightGBM) per range; positive values indicate MTGBM performed worse than LightGBM.

**Fig 6. Comparative performance summary between LightGBM and MTGBM. (A)** MSE comparison for CL and VD targets. **(B)** Test set prediction scatter plot for CL. **(C)** Test set prediction scatter plot for VD. **(D)** Residual plot for CL predictions. **(E)** Residual plot for VD predictions. **(F)** Model performance summary. Blue: LightGBM; red: MTGBM. Dashed lines indicate perfect prediction.

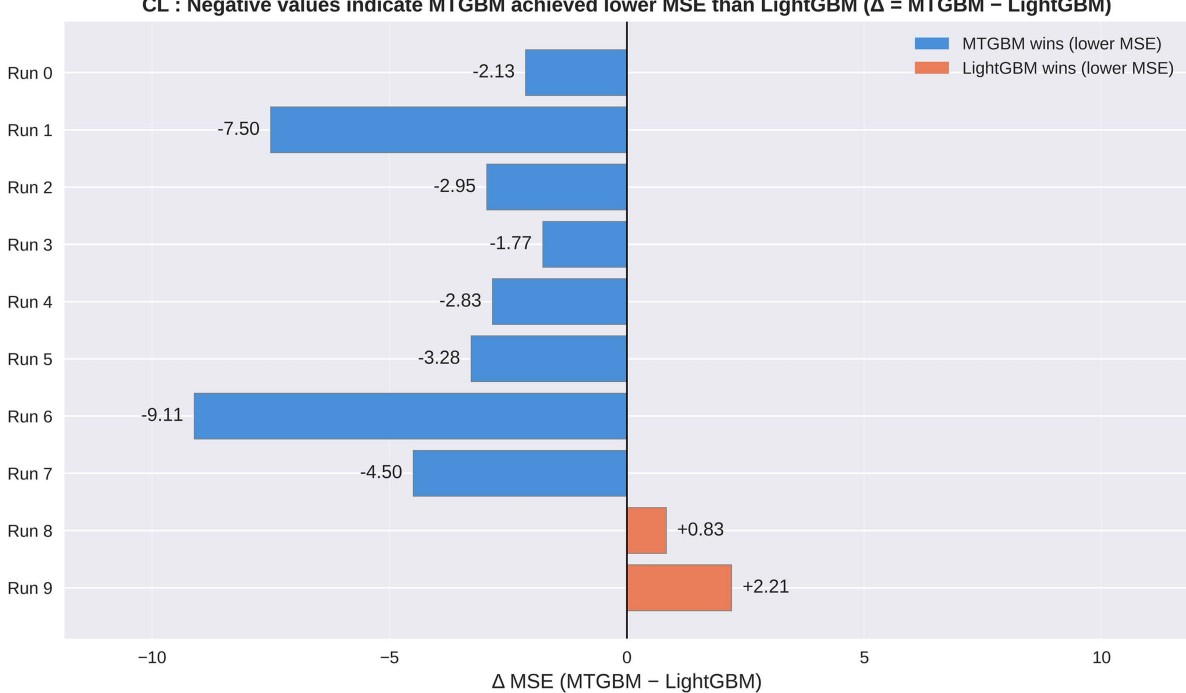

**Fig 7. Per-run CL MSE differences across 10 repeated random splits.** Horizontal bar plot of Δ MSE (MTGBM − LightGBM) for each of 10 independent data partitions. Blue bars indicate runs where MTGBM achieved lower MSE; orange bars indicate runs where LightGBM achieved lower MSE.

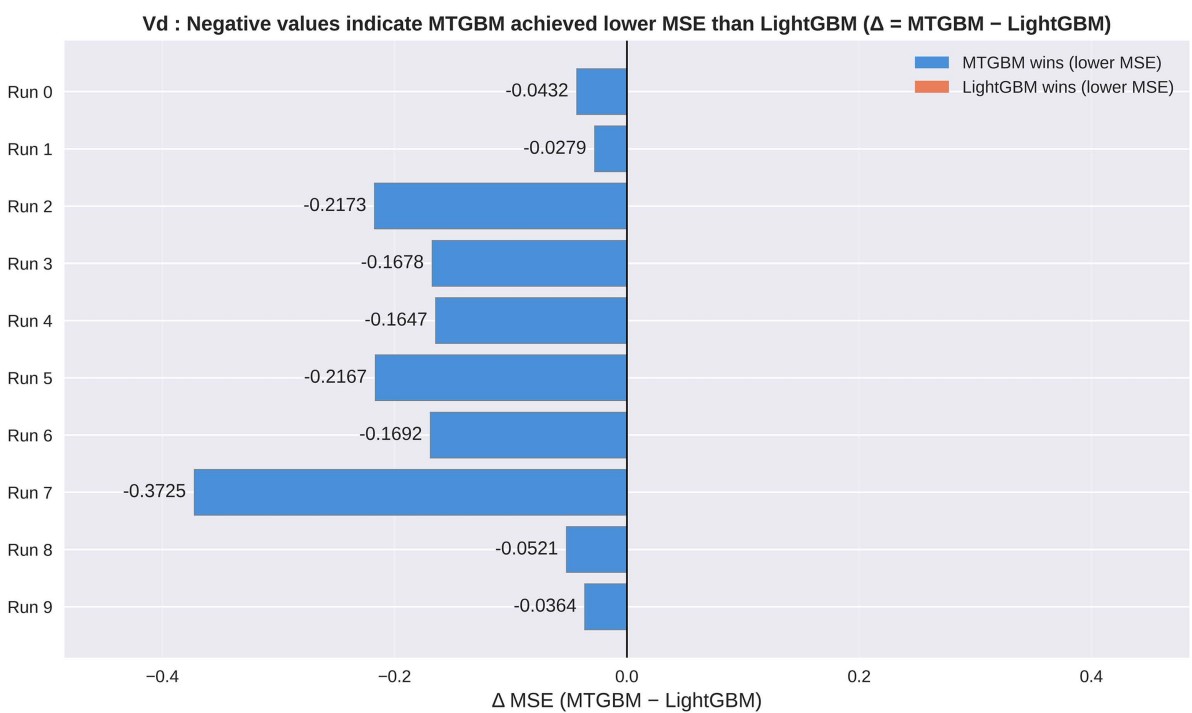

**Fig 8. Per-run VD MSE differences across 10 repeated random splits.** Horizontal bar plot of Δ MSE (MTGBM − LightGBM) for each of 10 independent data partitions. Blue bars indicate runs where MTGBM achieved lower MSE; orange bars indicate runs where LightGBM achieved lower MSE.

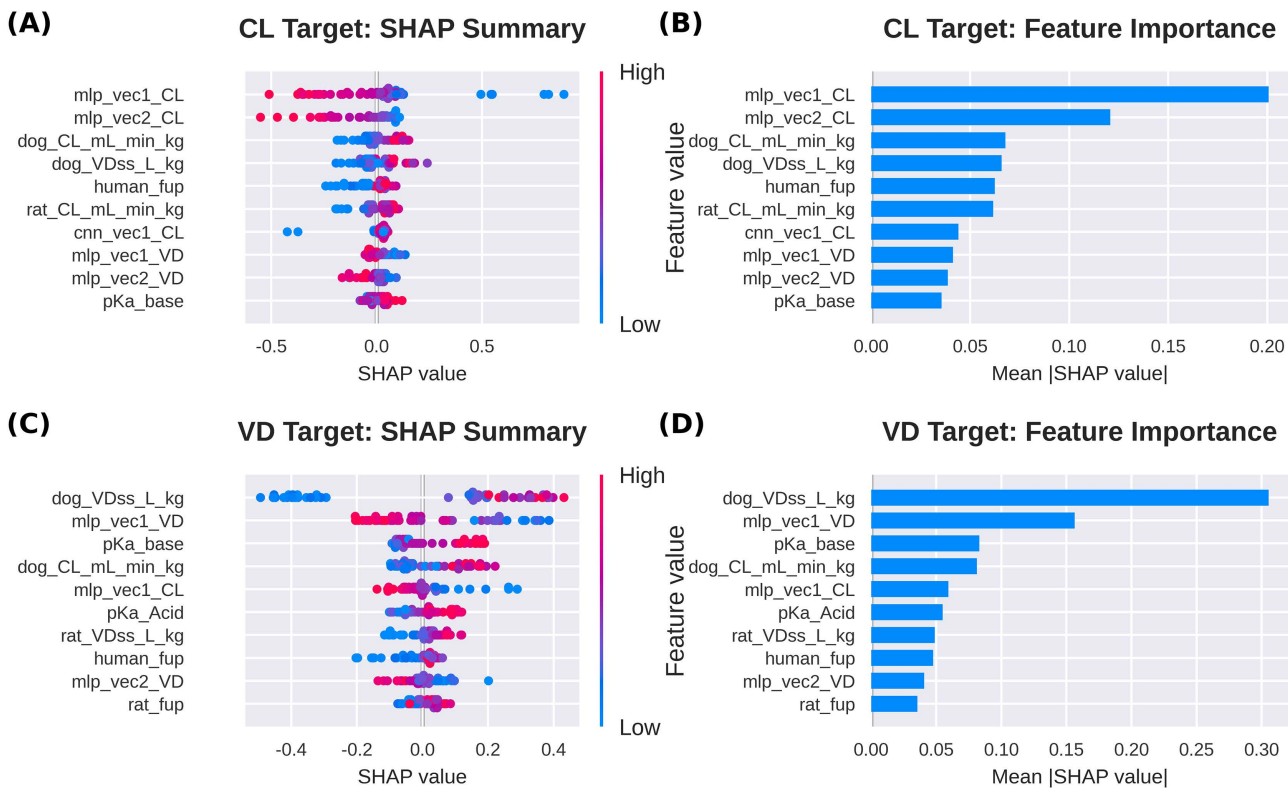

**Fig 9. SHAP summary plots for MTGBM. (A)** Beeswarm plot of SHAP values for top-10 features for CL prediction. **(B)** Bar plot of mean absolute SHAP values for CL prediction. **(C)** Beeswarm plot of SHAP values for top-10 features for VD prediction. **(D)** Bar plot of mean absolute SHAP values for VD prediction. Each point in beeswarm plots represents one test compound, colored by feature value (red: high; blue: low).

## 4.6. Overall assessment

The multi-task gradient boosting framework leveraged shared representations between CL and VD predictions, achieving generally favorable performance across multiple evaluation criteria. MTGBM outperformed single-task models on 5 out of 6 primary metrics, with degradation only in VD GMFE. The directional improvements, combined with integration of multi-modal molecular representations, suggest that MTGBM may offer practical benefits for simultaneous pharmacokinetic parameter prediction, pending validation on larger datasets with dedicated hyperparameter optimization.

## 5. Discussion and conclusion

This study demonstrates that a Multi-Task Gradient Boosting Machine (MTGBM) framework, integrating three complementary molecular modalities within shared decision trees, can achieve statistically significant improvements over single-task LightGBM baselines for simultaneous prediction of drug clearance and volume of distribution. MSE reductions of 23.6% for CL and 33.0% for VD, coupled with $R^2$ improvements of 40.0% and 97.2% respectively, suggest that jointly learning correlated pharmacokinetic endpoints captures structure-property relationships more effectively than independent single-task modeling. The robustness of these gains was confirmed across 10 repeated random splits, in which MTGBM achieved lower MSE in 8 out of 10 runs for CL and all 10 runs for VD, indicating that the primary single-split results reflect a genuine directional tendency rather than a favorable data partition. It should be noted, however, that the absolute $R^2$ values for CL remain modest (LightGBM: 0.140; MTGBM: 0.196), consistent with the well-recognized difficulty of CL prediction on small datasets, where multiple overlapping elimination pathways and high

inter-individual variability inherently limit the variance explained by structure-based models — a pattern reported in prior ML-based CL studies on similar datasets [2,3].

The differential improvement between CL and VD is pharmacokinetically interpretable. Volume of distribution, being more directly governed by physicochemical properties, may benefit more substantially from shared multi-modal representations. SHAP analysis revealed that MLP embeddings accounted for approximately 54% of top-10 feature importance for CL and 28% for VD, while preclinical PK parameters dominated VD prediction (47.8%), with MLP embeddings playing a strong complementary role. Cross-target features appeared in both directions — VD-specific embeddings ranked among the top-10 CL predictors, and the CL-specific embedding mlp_vec1_CL appeared at rank 5 among VD predictors — a pattern consistent with the physiological interdependence of CL and VD. An important caveat applies, however: since the CNN and MLP embeddings were trained directly on CL and VD as target variables, their prominence in SHAP analysis may partly reflect target-specific supervised training rather than solely multi-task information sharing. This limitation is inherent to supervised dimensionality reduction approaches rather than specific to our framework — conceptually analogous to Partial Least Squares (PLS) regression [20,21], where dimensionality reduction is performed along directions that maximize covariance with the response rather than variance in the predictors alone. The CNN and MLP embeddings thus represent a nonlinear extension of this well-established chemometric principle [20,21], and the utility of supervised multi-task deep learning for shared feature extraction across correlated ADME endpoints has been empirically demonstrated [22]. Complete disentanglement of supervised representation learning from multi-task information sharing would require comparison against unsupervised embeddings — such as those from self-supervised pretraining — which we identify as a priority for future work.

The achieved GMFEs (CL: 2.306, VD: 2.286) are competitive with prior ML-based PK studies; our CL GMFE falls below the typical range of 2.5–2.7 [8,11], while for VD, Murad et al. [9] reported that 75% of predictions fell within 3-fold error on a comparable dataset. The lowest reported GMFEs on the Iwata dataset (~1.6–1.9) were achieved by Iwata et al. [2] using a single-task multimodal approach with missing-value imputation across a larger compound pool. Direct comparison is limited, as our framework simultaneously predicts both targets using only compounds with measurements for both CL and VD, imposing an additional multi-task constraint. Notably, the overall VD GMFE increased by 13.5% despite MSE improving by 33.0%, a divergence driven exclusively by compounds in the low VD range (<0.5 L/kg, $\Delta = +2.274$), where GMFE is highly sensitive to relative fold errors. In contrast, MTGBM demonstrated superior GMFE in both mid and high VD ranges, confirming that the deterioration does not reflect uniform performance degradation. Both MSE and GMFE capture complementary aspects of prediction error, and their divergence underscores the importance of reporting multiple metrics rather than relying on a single criterion.

Key limitations of this study include the small dataset size (~700 compounds), which constrains statistical power — though the Diebold–Mariano test nonetheless reached significance for both CL ($p = 0.019$) and VD ($p = 0.004$). MTGBM hyperparameters were estimated from single-task results rather than optimized directly, due to the computational cost of the custom multi-task objective function; this may either underestimate MTGBM's true potential or overestimate it depending on how well single-task estimates approximate the multi-task optimum. The slightly higher CL standard deviation under MTGBM across repeated splits (3.373 vs 3.251) is consistent with this hyperparameter uncertainty, though VD prediction remained consistently superior across all 10 runs regardless of partition. Additionally, the supervised nature of the CNN and MLP embeddings limits the interpretability of SHAP-based cross-target attribution, as discussed above.

Future work should prioritize validation on external datasets from diverse chemical spaces, dedicated MTGBM hyperparameter optimization, and comparison against unsupervised molecular representations to isolate the contribution of multi-task learning from supervised feature extraction. Incorporating additional ADME endpoints and quantifying prediction uncertainty would further strengthen the framework's practical utility. Given that MTGBM demonstrated consistent improvements even with compact task-specific embeddings, it is plausible that stronger pretrained representations — such as those from foundation models trained on millions of molecules — would further amplify these gains.

In conclusion, MTGBM with multi-modal molecular representations establishes a proof-of-concept for simultaneous pharmacokinetic prediction, demonstrating that multi-task gradient boosting with complementary molecular modalities can yield statistically significant and robust improvements over single-task baselines. Broader applicability to drug discovery pipelines will depend on validation with larger datasets and dedicated hyperparameter optimization, but the results presented here provide a practical empirical foundation for future development of multi-modal, multi-task ADME modeling frameworks.

## Author contributions

**Conceptualization:** Gyu-Seong Lee.

**Formal analysis:** Gyu-Seong Lee.

**Funding acquisition:** Hyo Young Kim.

**Methodology:** Dae-Keun Park.

**Project administration:** Hyo Young Kim.

**Resources:** Hyeong-Gyu Choi, Hyo Young Kim.

**Software:** Dae-Keun Park.

**Supervision:** Hyo Young Kim.

**Validation:** Hyeong-Gyu Choi.

**Visualization:** Hyeong-Gyu Choi.

**Writing – original draft:** Hyo Young Kim.

**Writing – review & editing:** Hyo Young Kim.

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
