## [Decision Letter · Decision Letter 0]

17 Feb 2026

Dear Dr. Kim,

Thank you for submitting your manuscript to PLOS ONE. After careful consideration, we feel that it has merit but does not fully meet PLOS ONE’s publication criteria as it currently stands. Therefore, we invite you to submit a revised version of the manuscript that addresses the points raised during the review process.

We look forward to receiving your revised manuscript.

Kind regards,

Zixiang Gao

Academic Editor

PLOS One

Reviewers' comments:

Reviewer's Responses to Questions

**Comments to the Author**

1. Is the manuscript technically sound, and do the data support the conclusions?

Reviewer #1: Partly

Reviewer #2: Partly

2. Has the statistical analysis been performed appropriately and rigorously?

Reviewer #1: Yes

Reviewer #2: No

3. Have the authors made all data underlying the findings in their manuscript fully available?

Reviewer #1: Yes

Reviewer #2: No

4. Is the manuscript presented in an intelligible fashion and written in standard English?

Reviewer #1: Yes

Reviewer #2: No

Reviewer #1: This manuscript proposes a multi-task gradient boosting framework (MTGBM) that integrates multi-modal molecular representations to jointly predict human clearance (CL) and volume of distribution (Vd). The topic is timely and relevant, as accurate early-stage PK prediction remains a major bottleneck in drug discovery. The authors combine CNN-derived molecular image embeddings, descriptor- and fingerprint-based MLP embeddings, BorutaSHAP feature selection, and a custom multi-output gradient boosting objective. While the study is carefully executed and technically sound, its impact is substantially limited by presentation issues, modest empirical gains, and an overstatement of novelty relative to existing machine-learning literature. Overall, the work may be suitable for a general-scope journal, but it requires significant revision in presentation, positioning, and contextual discussion to meet current standards.

The presentation quality is a serious weakness. Many figures use extremely small fonts that are difficult to read even when zoomed in, particularly the BorutaSHAP plots and scatter plots comparing single-task and multi-task models. Given that interpretability (via SHAP) is emphasized as a key contribution, the unreadable axis labels, legends, and annotations significantly undermine the significance of the findings. This is not a cosmetic issue: poor readability directly affects the readers' ability to assess the claims. All figures should be regenerated with substantially larger fonts, simplified legends, and clearer layouts. In addition, some figures appear redundant (e.g., multiple scatter plots across splits conveying similar information) and could be consolidated to improve clarity.

The claimed novelty of the work is overstated. Multi-task gradient boosting machines with shared tree structures are not new and have already been formally introduced and analyzed. Moreover, from a practical standpoint, multi-output or multi-task tree-based models are well supported in widely used ecosystems such as LightGBM and scikit-learn, either directly or through standard wrappers. While it is true that such approaches are less commonly applied in PK prediction, this constitutes an application-level novelty rather than a methodological one.

Similarly, the construction of 'multi-task gradient boosting trees' in this manuscript largely follows established paradigms and does not introduce fundamentally new optimization strategies, regularization schemes, or task-coupling mechanisms. The improvements reported are modest (4.9% MSE reduction for CL and 10.8% for Vd) and, importantly, not statistically significant under the Diebold–Mariano test. As such, the authors should significantly temper their novelty claims and reframe the contribution as a careful application and validation study rather than a methodological advance.

A major omission is the lack of discussion and comparison with recent blind-prediction challenges and benchmarks that strongly support the superiority of multi-modal, multi-task pretraining regimes. For example, recent work reported in Journal of Chemical Information and Modeling 65 (19), 10465-10476 demonstrates that large-scale, multi-modal, multi-task pretraining consistently outperforms task-specific models on ADME-relevant endpoints, especially in low-data regimes. Conceptually, the current MTGBM framework addresses a similar motivation but does so in a comparatively shallow, non-pretrained setting.

The authors should explicitly discuss how their approach relates to, differs from, and potentially complements such pretraining-based frameworks. In particular, it would be valuable to clarify whether the modest gains observed here stem from limited dataset size (~700 compounds) and whether the approach would scale or remain competitive in settings where pretrained representations are available.

Relatedly, the manuscript does not engage with the rapidly growing body of work on large models and foundation models for molecular and drug discovery. Recent studies (e.g., arXiv:2511.11257) have demonstrated that large, pretrained models (often trained across millions of molecules and multiple objectives) can achieve outstanding predictive power on downstream PK and ADME tasks, frequently surpassing classical ML pipelines even without extensive feature engineering.

Given this context, the authors' reliance on relatively small CNN and MLP embeddings trained specifically for CL and Vd should be critically discussed. Are the gains from MTGBM expected to persist when stronger, pretrained representations are used? Could the proposed multi-task boosting layer act as a lightweight downstream learner on top of foundation-model embeddings? Without such discussion, the work risks appearing disconnected from the current trajectory of the field.

Reviewer #2: This study presents a novel approach to predicting key pharmacokinetic parameters by combining multi-modal molecular representations with multi-task learning. The authors have clearly put significant thought into designing their MTGBM framework, and the integration of different feature types including CNN-based structural embeddings, molecular fingerprints, and physicochemical descriptors represents a thoughtful attempt to capture complementary information about drug molecules. The use of BorutaSHAP for feature selection is also a nice touch that adds rigor to the modeling process.

However, I do have some concerns about the work that should be addressed before publication. The most significant issue is the lack of statistical significance in the improvements over single-task models. While the authors acknowledge this limitation, they somewhat overstate the potential of their approach throughout the manuscript. The Diebold-Mariano test results with p-values of 0.117 and 0.245 for CL and VD respectively indicate that we cannot rule out the possibility that the observed improvements occurred by chance. Given that the dataset contains only about 700 compounds, this is perhaps not surprising, but the discussion should more carefully frame these findings as preliminary evidence rather than definitive proof of the method's superiority.

The dataset size itself poses another challenge. With only around 700 compounds split into training, validation, and test sets, the test set ends up being quite small at 139 samples. This raises questions about the stability of the performance metrics and whether they might be heavily influenced by a few outliers. The authors mention using a robust optimization algorithm for splitting, but it would be helpful to know more about how stable the results were across different random splits.

I also have some methodological concerns about the feature extraction process. The authors trained separate CNN and MLP models for CL and VD predictions before extracting embeddings for the MTGBM model. This creates a potential information leakage issue since these embedding models were trained using the same data that later gets used for training and evaluating the MTGBM. Ideally, the embedding extraction should be done in a way that keeps test data completely separate, perhaps through nested cross-validation or by training the embedding models only on training data before any validation or testing occurs.

The hyperparameter optimization approach for the MTGBM model is another area that needs clarification. The authors mention using "estimated optimal parameters" from single-task models due to computational constraints. While this is understandable, it does mean we cannot be sure that the MTGBM model is performing at its full potential. A properly optimized multi-task model might show larger improvements, or conversely, the current results might be underestimating its true performance. Either way, this limitation should be more prominently discussed.

The interpretability analysis using SHAP values raises some questions about what exactly the MLP and CNN embeddings represent. These are learned representations from neural networks trained specifically to predict CL and VD, so it is somewhat circular when they emerge as important features in the SHAP analysis. The authors interpret this as evidence of successful cross-task information sharing, but it might simply reflect that these embeddings already contain target-specific information from their training process. A cleaner approach might have been to use unsupervised molecular embeddings that were not trained on the prediction tasks themselves.

I noticed some inconsistencies in the reported results that need attention. For VD prediction, the MSE improved substantially from 0.950 to 0.848, yet the GMFE increased from 1.844 to 1.938. The authors suggest this indicates the importance of evaluating multiple metrics, which is valid, but this pattern deserves more discussion. Typically, improvements in MSE would be expected to correspond with better GMFE, so the divergence here might indicate something interesting about the error distribution that could be explored further.

The manuscript could also benefit from more careful editing in places. The literature review section repeats some content between sections 2.1 and 2.2, and there are minor formatting inconsistencies in the references. The figures mentioned in the text, particularly Figures 1 through 6, are referenced but not actually included in the manuscript file, making it impossible to fully evaluate the visual presentations of the results.

Despite these concerns, the work makes a valuable contribution by demonstrating how multi-task learning can be applied to pharmacokinetic prediction. The idea of leveraging correlations between related parameters is conceptually sound and worth exploring further. The authors have been transparent about their limitations, which is commendable. With some revisions to address the methodological issues and a more nuanced interpretation of the results, this could become a useful addition to the literature on computational pharmacokinetics. The framework they have developed could potentially be extended to other ADME properties, and their careful documentation of the modeling process provides a good foundation for others building on this work.

.

Reviewer #1: **Yes:** Zhaoxi SunZhaoxi SunZhaoxi SunZhaoxi Sun

Reviewer #2: No

---

## [Author Response · Author response to Decision Letter 1]

3 Apr 2026

Thank you for your careful review of our manuscript. We have addressed all comments raised by both reviewers in detail. Our point-by-point responses to each comment, along with a description of all changes made to the manuscript, are provided in the accompanying "Response to Reviewers" document. We hope the revised manuscript meets the standards for publication in PLOS ONE and look forward to your feedback.

---

## [Decision Letter · Decision Letter 1]

13 Apr 2026

Multi-Task Gradient Boosting with Multi-Modal Molecular Representations for Simultaneous Prediction of Drug Clearance and Volume of Distribution

PONE-D-25-68444R1

Dear Dr. Kim,

We’re pleased to inform you that your manuscript has been judged scientifically suitable for publication and will be formally accepted for publication once it meets all outstanding technical requirements.

Kind regards,

Zixiang Gao

Academic Editor

PLOS One

Additional Editor Comments (optional):

Thank you for your patience and for carefully addressing the reviewers' comments. We are pleased to accept your manuscript for publication in PLOS ONE.

Reviewers' comments:

Reviewer's Responses to Questions

**Comments to the Author**

Reviewer #1: All comments have been addressed

Reviewer #2: All comments have been addressed

2. Is the manuscript technically sound, and do the data support the conclusions?

Reviewer #1: Partly

Reviewer #2: Yes

3. Has the statistical analysis been performed appropriately and rigorously?

Reviewer #1: Yes

Reviewer #2: Yes

4. Have the authors made all data underlying the findings in their manuscript fully available?

Reviewer #1: Yes

Reviewer #2: (No Response)

5. Is the manuscript presented in an intelligible fashion and written in standard English?

Reviewer #1: Yes

Reviewer #2: Yes

Reviewer #1: (No Response)

Reviewer #2: Accept

Accept

Accept

Accept

Accept

Accept

AcceptAcceptAcceptAcceptAcceptAcceptAcceptAcceptAcceptAccept

.

Reviewer #1: **Yes:** Zhaoxi SunZhaoxi SunZhaoxi SunZhaoxi Sun

Reviewer #2: No

---

## [Editor Report · Acceptance letter]

PONE-D-25-68444R1

PLOS One

Dear Dr. Kim,

I'm pleased to inform you that your manuscript has been deemed suitable for publication in PLOS One. Congratulations! Your manuscript is now being handed over to our production team.

Kind regards,

on behalf of

Dr. Zixiang Gao

Academic Editor

PLOS One